# Incretin-Based Multi-Agonist Peptides Are Neuroprotective and Anti-Inflammatory in Cellular Models of Neurodegeneration

**DOI:** 10.3390/biom14070872

**Published:** 2024-07-19

**Authors:** Katherine O. Kopp, Yazhou Li, Elliot J. Glotfelty, David Tweedie, Nigel H. Greig

**Affiliations:** 1Translational Gerontology Branch, Intramural Research Program, National Institute on Aging, National Institutes of Health, Baltimore, MD 21224, USA; katieokopp@gmail.com (K.O.K.); liyaz@mail.nih.gov (Y.L.); tweedieda@grc.nia.nih.gov (D.T.); 2Cellular Stress and Inflammation Section, Intramural Research Program, National Institute on Drug Abuse, National Institutes of Health, Baltimore, MD 21224, USA; elliot.glotfelty@nih.gov

**Keywords:** glucagon-like peptide-1 (GLP-1), glucose-dependent insulinotropic polypeptide (GIP), glucagon, incretin mimetic, neuroinflammation, neurodegeneration, Alzheimer’s disease, Parkinson’s disease, microglia

## Abstract

Glucagon-like peptide-1 (GLP-1)-based drugs have been approved by the United States Food and Drug Administration (FDA) and are widely used to treat type 2 diabetes mellitus (T2DM) and obesity. More recent developments of unimolecular peptides targeting multiple incretin-related receptors (“multi-agonists”), including the glucose-dependent insulinotropic polypeptide (GIP) receptor (GIPR) and the glucagon (Gcg) receptor (GcgR), have emerged with the aim of enhancing drug benefits. In this study, we utilized human and mouse microglial cell lines, HMC3 and IMG, respectively, together with the human neuroblastoma SH-SY5Y cell line as cellular models of neurodegeneration. Using these cell lines, we studied the neuroprotective and anti-inflammatory capacity of several multi-agonists in comparison with a single GLP-1 receptor (GLP-1R) agonist, exendin-4. Our data demonstrate that the two selected GLP-1R/GIPR dual agonists and a GLP-1R/GIPR/GcgR triple agonist not only have neurotrophic and neuroprotective effects but also have anti-neuroinflammatory properties, as indicated by the decreased microglial cyclooxygenase 2 (COX2) expression, nitrite production, and pro-inflammatory cytokine release. In addition, our results indicate that these multi-agonists have the potential to outperform commercially available single GLP-1R agonists in neurodegenerative disease treatment.

## 1. Introduction

Significant scientific, medical, and media attention has been given to incretin mimetics in the treatment of type 2 diabetes mellitus (T2DM) and obesity, as well as in numerous additional seemingly unrelated conditions—and rightfully so. Glucagon-like peptide-1 (GLP-1) and, more recently, glucose-dependent insulinotropic polypeptide (GIP) and glucagon (Gcg) receptor (R) agonists have revolutionized T2DM and obesity treatment, restoring insulin sensitivity and often facilitating weight loss in patients [1,2]. Moreover, current research suggests the potential for repurposing this drug class to treat various additional diseases, including prostate cancer [3], polycystic ovary syndrome [4], substance abuse [5], and diseases/disorders of the brain with an inflammatory component, such as traumatic brain injury (TBI) [6,7], ischemic stroke [8,9], idiopathic intracranial hypertension [10], and neurodegenerative diseases including Alzheimer’s (AD) [11,12] and Parkinson’s diseases (PD) [8,11,13]. Given the insufficiency of existing treatment options for these disorders, particularly AD and PD, repurposing already United States Food and Drug Administration (FDA)-approved incretin mimetics could yield a novel therapeutic strategy to more efficiently meet the urgent need to treat neurodegenerative diseases.

A plethora of epidemiological studies have established T2DM as a significant risk factor for dementia and neurodegenerative disease [14,15,16,17,18]. Further studies have found that T2DM patients who were treated with GLP-1R agonists were at reduced risk for developing AD or PD [19,20] and were protected against stroke [21,22]. Indeed, brain insulin resistance (IR), chronic neuroinflammation, and oxidative stress are hallmarks and driving forces of neurodegenerative disease [23,24,25], which current treatment modalities for neurodegeneration neglect to address [26]. Seminal preclinical studies beginning in 2002 first illustrated the neurotrophic and neuroprotective capacity of GLP-1R agonists—GLP-1 and structurally related GLP-1R agonist exendin-4 were found to enhance neurite outgrowth, promote nerve growth factor (NGF)-mediated neuronal differentiation, protect against apoptosis, and maintain cholinergic neuronal health and function in cultured neurons and rodent models of neurodegeneration [27,28,29]. For the next two decades, our group and numerous others continued to preclinically investigate molecular pathways conferring the neuroprotective and anti-neurodegenerative benefits of incretin mimetics [30,31,32,33,34,35], as well as testing numerous formulations of incretin-based receptor agonists in neurodegenerative disease models [36,37,38,39].

In the human central nervous system (CNS), incretin mimetics have the potential to enact insulinotropic, neuroprotective, antiapoptotic, antioxidative, and anti-inflammatory benefits—pleiotropic mechanisms of action that can afford various anti-neurodegenerative protections. Clinical trials investigating commercially available GLP-1R agonists as treatments for PD and AD have already been completed, with encouraging results comprising improved off-medication motor scores, motor ability, and cognitive function in PD patients [40,41,42,43], as well as improved brain glucose metabolism [44] and reduced amyloid-β 1-42 concentration in neuronal-derived extracellular vesicles [45] in AD patients. Numerous ongoing clinical trials are investigating various GLP-1R agonist analogs, including exenatide, liraglutide, semaglutide, and lixisenatide, as anti-neurodegenerative agents in AD and PD patients [46,47].

Recently, the development of unimolecular incretin receptor multi-agonists has received much attention in the T2DM field. Tirzepatide/*Mounjaro*^®^, a novel GLP-1R/GIPR dual agonist, was recently approved by the FDA for T2DM treatment, and demonstrated enhanced insulinotropic benefits relative to single GLP-1R agonists in clinical trials [48]. The development and evaluation of additional GLP-1R/GIPR dual and GLP-1R/GIPR/GcgR triple agonists are well underway for T2DM treatment, but studies in repurposing them for neurodegenerative disorders lag behind. In this work, we aim to aid the evaluation of these novel incretin receptor multi-agonists in preclinical models of neurodegeneration and thereby inform potential future clinical trials. We highlight promising findings in cell culture models of neurodegeneration indicating the enhanced neuroprotective, antioxidative, and anti-inflammatory benefits of the three selected multi-agonists—the dual GLP-1R/GIPR agonists “Twincretin” and LY329 (known commercially as Tirzepatide/*Mounjaro^®^*) and the triple GLP-1R/GIPR/GcgR agonist “Triagonist” (known clinically as Retatrutide)—relative to a single GLP-1R agonist, exendin-4 (known clinically as Exenatide).

## 2. Materials and Methods

### 2.1. Materials

The GLP-1R agonists GLP-1 and exendin-4 were purchased and three multi-agonists (two dual GLP-1R/GIPR agonists, Twincretin and LY3298176 (Tirzepatide), and a triple GLP-1R/GIPR/GcgR agonist, LY3437943 (Retatrutide)) were obtained through custom synthesis from AnaSpec, Inc. (Fremont, CA, USA). A 30% hydrogen peroxide solution (*w*/*w*) in H_2_O (H_2_O_2_) was purchased from Sigma-Aldrich Corporation (H1009, St. Louis, MO, USA). Staurosporine (STS) was obtained from TOCRIS BioScience (1285, Minneapolis, MN, USA). Lipopolysaccharide (LPS) was purchased from Sigma-Aldrich Corporation (L4524) and prepared with phosphate-buffered saline (PBS) as a stock of 2.5 mg/mL.

### 2.2. Cell Culture

All cell line cultures were maintained in a 37 °C incubator comprising 5% CO_2_ and 95% air, with medium replacement every other day.

#### 2.2.1. SH-SY5Y Neuroblastoma Cells

Human SH-SY5Y neuronal cells were purchased from American Type Culture Collection (ATCC) (CRL-2266, Manassas, VA, USA) and grown in a mixture of half Eagle’s minimum essential medium (EMEM) (30-2003, ATCC) and half Ham’s F12-K (Kaighn’s) medium (30-2004, ATCC) supplemented with 10% heat-inactivated fetal bovine serum and 100 U/mL penicillin/streptomycin (15140122, Invitrogen, Carlsbad, CA, USA). Cells were split at a 1:3 ratio every 7 days using 0.25% trypsin-EDTA (25200056, Invitrogen). A maximum of 15 passages were used for these cells.

#### 2.2.2. HMC3 Microglia

The transformed human microglial cell line HMC3 was also obtained from ATCC (CRL-3304), retaining many of the properties of primary microglial cells. HMC3 cells were cultured with EMEM (30-2003, ATCC) supplemented with 10% heat-inactivated fetal bovine serum and 100 U/mL penicillin/streptomycin (15140122, Invitrogen). Cells were split at a 1:5 ratio every 7 days using 0.25% trypsin-EDTA (25200056, Invitrogen). A maximum of 10 passages were used for these cells.

#### 2.2.3. Immortalized Mouse Microglia (IMG)

The immortalized mouse microglia (IMG) cell line was obtained from Millipore (SCC134, Burlington, MA, USA) and used as an additional neuroinflammation model [49,50]. IMG cells were cultured in high-glucose Dulbecco’s modified Eagle medium (DMEM, D6546, Sigma-Aldrich) with 1X Gibco GlutaMAX^TM^ supplement (35050-061, Invitrogen), 10% heat-inactivated fetal bovine serum, and 100 U/mL penicillin/streptomycin (15140122, Invitrogen). Cells were split at a 1:10–1:20 ratio every 5 days using a cell scrape. A maximum of 10 passages were used for these cells.

### 2.3. Drug and Challenge Dose Selection

The selection of an incretin mimetic dose for evaluation in neurotrophic, neuroprotective, and inflammatory studies across SH-SY5Y, HMC3, and IMG cells was based on pilot dose–response studies that were focused on identifying the threshold concentration to provide the biological action of interest, and this dose was then used across single, dual, and triple receptor agonists to allow their side-by-side comparison at an equimolar concentration. Likewise, for the selection of a physiological/drug-induced (i.e., H_2_O_2_, LPS, STS) challenge, concentrations were selected from prior pilot dose–response studies focused to achieve a biologically significant but sub-maximal effect.

### 2.4. Cell Viability Assays

Cell viability was assessed using either the CytoTox-ONE™ Homogeneous Membrane Integrity Assay (G7891, Promega, Madison, WI, USA) or the CellTiter 96^®^ Aqueous One Solution Cell Proliferation Assay kit (MTS) (G3581, Promega). Assay plates were read using an Infinite m200 Pro plate reader with i-Control software (Tecan i-control 1.11.1.0, Tecan, Morrisville, NC, USA).

### 2.5. Cellular ROS Assay

The Fluorescent DCFDA Cellular Reactive Oxygen Species (ROS) Detection Assay was obtained from Abcam (ab113851, Waltham, Boston, MA, USA). SH-SY5Y and HMC3 cells were seeded at a density of 2.5 × 10^4^ cells per well the day before treatment. To induce oxidative stress in both cell lines, we used a 1 h H_2_O_2_ treatment. Prior to the H_2_O_2_ challenge, cells were pre-treated with 100 nM incretin-based peptide or vehicle for 2 h. Cellular ROS were measured with the assay kit in accordance with the manufacturer’s protocol. Assay plates were read using an Infinite m200 Pro plate reader with i-Control software (Tecan).

### 2.6. Cytokine ELISAs

IMG cells were seeded in 24-well dishes at a density of 2.0 × 10^5^ cells per well. Following a 24 h 100 nM incretin-based peptide pre-treatment and a 24 h 1 ng/mL LPS pro-inflammatory stimulus, cell media was collected for use in cytokine and prostaglandin enzyme-linked immunosorbent assays (ELISAs), as well as in Griess nitrite assays. Mouse interleukin (IL)-6 and mouse tumor necrosis factor-alpha (TNF-α) levels in cell culture media samples were measured utilizing BioLegend’s corresponding ELISA MAX™ Deluxe Sets (431315 and 430915, San Diego, CA, USA) in accordance with the manufacturer’s protocol. Assay plates were read using a SpectraMax Plus 384 Microplate Reader with SoftMax^®^ Pro 7 software (Molecular Devices, Sunnyvale, CA, USA).

### 2.7. Prostaglandin E_2_ ELISAs

The Prostaglandin E_2_ ELISA Kit–Monoclonal was purchased from Cayman Chemical (514010, Ann Arbor, MI, USA) and utilized to quantify prostaglandin E_2_ (PGE2) levels in undiluted cell culture media samples obtained as indicated above in accordance with the manufacturer’s protocol. Assay plates were read using an Infinite m200 Pro plate reader with i-Control software (Tecan).

### 2.8. Nitrite Detection Assay

The Griess Reagent System, a colorimetric nitrite assay kit, was purchased from Promega (G2930) and utilized to quantify nitrite levels in undiluted cell culture media samples obtained as indicated above in accordance with the manufacturer’s protocol. Assay plates were read using a SpectraMax Plus 384 Microplate Reader with SoftMax^®^ Pro 7 software (Molecular Devices).

### 2.9. Caspase-3 Activity Assay

The Fluorimetric Caspase-3 Assay Kit was purchased from Sigma (CASP3F-1KT, Saint Louis, MO, USA). SH-SY5Y cells were seeded in 24-well plates at a density of 1.25 × 10^5^ cells per well the day before treatment. Cells were first pre-treated with incretin-based peptide for 24 h, followed by the addition of STS to its final concentration for another 24 h. At the end of treatment, cells were collected in lysis buffer and spun at 10,000× *g* for 5 min to remove debris. Supernatant was used in the caspase-3 activity assay as well as a BCA assay for the measurement of protein content and used in accordance with the manufacturer’s protocol. Assay plates were read using an Infinite m200 Pro plate reader with i-Control software (Tecan), and the final caspase-3 activity results were normalized to protein content in the samples.

### 2.10. Western Blot

IMG cells were seeded in 60 mm dishes at a density of 2 × 10^6^ cells per dish. After a 24 h 100 nM incretin-based peptide pre-treatment and a 24 h 1 ng/mL LPS pro-inflammatory stimulus, cells were harvested using a cell scrape, collected by centrifugation, and re-suspended in RIPA buffer (R0278, Sigma-Aldrich) with 1X Halt^TM^ Protease & Phosphatase Single-Use Inhibitor Cocktail (78442, ThermoFisher Scientific, Waltham, MA, USA) for lysis. Samples were vortexed heavily every 5 min for 30 min and centrifuged at the end of the lysis period. Supernatant was collected for Western blotting and the Pierce BCA Protein Assay (23225, ThermoFisher Scientific) to determine total protein concentration.

Protein samples were resolved on NuPAGE^TM^ 4–12% Bis-Tris Gels (NP0336BOX, Invitrogen) in NuPAGE^TM^ MES SDS Running Buffer (NP0002, Invitrogen), along with a Chameleon^®^ Duo Pre-stained Protein Ladder (928-60000, LI-COR, Lincoln, NE, USA). Gels were transferred to Immobilon-FL PVDF Membranes (IPFL00010, Millipore) using a Pierce^TM^ Power Blotter (22834, ThermoFisher Scientific) semi-dry transfer system with 1-Step^TM^ Transfer Buffer (84731, ThermoFisher Scientific).

Membranes were blocked using Intercept^®^ Blocking Buffer (927-60001, LI-COR) and stained with primary antibodies for inflammatory marker cyclooxygenase 2 (COX2, 12282, Cell Signaling Technology) at a 1:1000 dilution as well as for housekeeping protein glyceraldehyde-3-phosphate dehydrogenase (GAPDH, MA5-15738, Invitrogen) at a 1:4000 dilution for normalization. The secondary antibody used for the visualization of GAPDH was IRDye^®^ 680RD Goat anti-Mouse (925-68070, LI-COR) at a 1:30,000 dilution; the secondary antibody used for the visualization of COX2 was IRDye^®^ 800CW Donkey anti-Rabbit (925-32213, LI-COR) at a 1:30,000 dilution. Images were obtained using an Odyssey CLx Near-Infrared Fluorescence Imaging System (LI-COR) paired with Image Studio^TM^ software (Image Stu-dio^TM^ Ver 5.2, LI-COR). Densitometry analysis was performed directly on Image Studio^TM^.

### 2.11. Statistical Analysis

Results are provided as the mean ± standard error of the mean values throughout. Statistical analyses, detailed in the legend of each figure, consisted of one-way ANOVA with either Dunnett’s or Tukey’s multiple comparisons test depending on the experiment, all with a significance threshold of α = 0.05. Statistical analyses were performed using GraphPad Prism software (GraphPad Prism 10.2.3, GraphPad Software, Boston, MA, USA).

## 3. Results

### 3.1. Trophic Properties of Exendin-4 and Multi-Agonists in Neuronal and Microglial Cell Lines

Our research group and others have established that single GLP-1R agonists can promote neurotrophic effects in human neuronal SH-SY5Y cells [30,51,52,53,54]. We now seek to compare the trophic abilities of the dual GLP-1R/GIPR agonists Twincretin and LY329 as well as the triple GLP-1R/GIPR/GcgR agonist Triagonist with those of the single GLP-1R agonist exendin-4 in SH-SY5Y cells as well as two microglial cell lines (Figure 1). To this end, we treated SH-SY5Y cells with 10 nM exendin-4, Twincretin, LY329, or Triagonist for 48 h and compared cell viability via MTS assays (Figure 1A). Each incretin-based peptide significantly enhanced SH-SY5Y viability relative to the untreated control (exendin-4 *p* < 0.0001, Twincretin *p* < 0.01, LY329 *p* < 0.05, Triagonist *p* < 0.0001). We performed a similar study in human microglial HMC3 cells—cells were treated with a 10 nM concentration of each incretin peptide and viability was determined via MTS assays after 48 h (Figure 1B). Only the dual GLP-1R/GIPR agonists, Twincretin and LY329, significantly enhanced HMC3 viability (*p* < 0.01). In contrast, in mouse IMG microglial cells treated for 48 h with 100 nM of each drug, exendin-4 (*p* < 0.001) and all multi-agonists (*p* < 0.0001) significantly enhanced IMG cell viability relative to the control (Figure 1C), with the multi-agonists enhancing it to a greater degree than exendin-4 (*p* < 0.0001). These incretin-based peptide concentrations were specifically selected from pilot study dose–response curves in each cell line focused on defining the threshold activity of the peptides.

### 3.2. Multi-Agonists Are Neuroprotective and Antioxidant against H_2_O_2_ Challenge in SH-SY5Y Neurons and HMC3 Microglia

We have previously shown the neuroprotective potential of incretin-based single and multi-agonists against a variety of cellular insults and CNS injuries in both cell culture and animal models of neurodegeneration [7,8,51,55,56]. In the present study, we directly compare the neuroprotective potential of single GLP-1R agonist exendin-4 vs. that of dual and triple incretin-based receptor agonists in both microglial and neuronal cell lines (Figure 2A,B). For HMC3 (Figure 2A) and SH-SY5Y (Figure 2B) cell viability assays, cells were pre-treated with vehicle or 10 nM exendin-4, Twincretin, LY329, or Triagonist for 24 h; then, a 400 µM H_2_O_2_ challenge—determined via pilot dose–response studies to achieve 25–50% cell death—was added for another 24 h to stimulate oxidative stress and neurotoxicity. Cell viability was determined via MTS assay. Under this condition, the multi-agonists Twincretin, LY329, and Triagonist significantly enhanced SH-SY5Y viability relative to the H_2_O_2_ challenge alone (*p* < 0.0001), whereas exendin-4 did not (Figure 2B). LY329 (*p* < 0.01) and Triagonist (*p* < 0.001) treatments significantly enhanced SH-SY5Y viability relative to exendin-4 treatment (Figure 2B). In HMC3 cells, all multi-agonists (*p* < 0.0001) and the single agonist exendin-4 (*p* < 0.01) significantly enhanced viability in the presence of H_2_O_2_, with Twincretin (*p* < 0.0001) and LY329 (*p* < 0.05) significantly outperforming exendin-4 (Figure 2A).

Our previous work additionally found that the GLP-1 metabolite GLP-1 (9–36) and exendin-4 were effective in significantly reducing ROS in SH-SY5Y cells [51]; we now test whether our multi-agonists of interest demonstrate similar antioxidative properties (Figure 2C,D). HMC3 microglial cells (Figure 2C) were pre-treated for 2 h with the vehicle or 100 nM GLP-1, Twincretin, LY329, or Triagonist, then exposed to either the vehicle or 150 µM H_2_O_2_ for 1 h. The 150 µM H_2_O_2_ concentration was determined in prior dose–response studies to achieve an approximate 50% elevation in ROS; the 100 nM incretin-based peptide concentration was selected from pilot study dose–response data focused on defining threshold activity. ROS levels were determined by a DCFDA assay. In the absence of H_2_O_2_, the multi-agonists Twincretin, LY329, and Triagonist significantly reduced basal ROS levels in HMC3 cells (Figure 2C; Twincretin *p* < 0.01, LY329 *p* < 0.0001, Triagonist *p* < 0.05); however, with H_2_O_2_ challenge, no significant incretin antioxidant effects were observed in HMC3 cells. SH-SY5Y neuronal cells (Figure 2D) were similarly pre-treated for 2 h with the vehicle or 100 nM Twincretin, LY329, or Triagonist and exposed to a 1 h 150 µM H_2_O_2_ challenge. Unlike in the HMC3 cells, LY329 and Triagonist significantly or near-significantly (respectively) reduced ROS in the presence of H_2_O_2_ in SH-SY5Y cells (Figure 2D; LY329 *p* < 0.05, Triagonist *p* = 0.0582).

### 3.3. Exendin-4 and Multi-Agonists Are Antiapoptotic in STS-Challenged SH-SY5Y Neurons

Neuronal and glial apoptosis is a driving feature of cognitive and physical impairment in neurodegenerative disease. In order to study the potential antiapoptotic effects of our incretin-based multi-agonists, we employed STS, a compound known to initiate apoptosis via caspase-3-dependent and -independent mechanisms [57], in human neuronal SH-SY5Y cells to simulate pro-apoptotic and neurotoxic conditions of the neurodegenerative disease brain. We optimized the STS dose and treatment time to achieve 50–150% elevations in cytotoxicity. Cells were first pre-treated with 100 nM concentrations of the single agonist exendin-4 (Figure 3A), the dual agonist Twincretin (Figure 3B), or the triple agonist Triagonist (Figure 3C) for 24 h, then were exposed to a 100 or 250 nM STS challenge for an additional 24 h. These incretin and STS doses were selected from prior pilot dose–response curves. Cytotoxicity, as measured by the CytoTox-ONE™ lactate dehydrogenase (LDH) assay, was dose-dependently elevated by STS, and was significantly reduced in both 100 and 250 nM STS conditions by exendin-4 (Figure 3A, *p* < 0.05) and by Twincretin (Figure 3B, *p* < 0.01). Triagonist significantly reduced cytotoxicity only in the 250 nM STS condition (Figure 3C, *p* < 0.0001).

To further investigate cellular mechanisms behind this observed antiapoptotic activity, we quantified caspase-3 activity in STS-challenged SH-SY5Y cells treated with our incretin-based peptides of interest. SH-SY5Y cells were pre-treated for 24 h with 100 nM concentrations of exendin-4, Twincretin, or Triagonist and were then challenged with 250 nM STS for 24 h. Interestingly, only the Triagonist treatment significantly reduced caspase-3 activity relative to STS challenge alone (Figure 4), suggesting that perhaps the triple agonist we used in this study may be superior to single or dual agonists in terms of reducing caspase-3 activity under this condition.

### 3.4. Multi-Agonists Dampen Neuroinflammatory Signaling of LPS-Stimulated IMG Cells

To study the potential for incretin receptor multi-agonists to mitigate neuroinflammation as an additional anti-neurodegenerative mechanism of action, we utilized mouse IMG cells challenged with LPS as a cellular model of neuroinflammation. IMG cells have been established as a useful microglial cell line for studies of neuroinflammation [50,58], and also have been shown to express GLP-1R and other incretin receptors [51]. LPS, a membrane component of Gram-negative bacteria, binds to Toll-like receptor 4 (TLR4) and stimulates downstream signaling pathways, culminating in microglial production and the release of inflammatory and neurotoxic factors, including pro-inflammatory cytokines, chemokines, and prostaglandins [59,60]. In early optimization experiments, we tested various doses and treatment times of LPS in IMG cells and found that 1 ng/mL LPS for 24 h produced a consistent elevation in inflammatory markers within detection range of our assays.

As in our previous experiments with HMC3 and SH-SY5Y cells, IMG cells were pre-treated for 24 h with the vehicle or 100 nM concentrations of exendin-4, Twincretin, LY329, or Triagonist. Cells were then exposed to a pro-inflammatory challenge of 1 ng/mL LPS for 24 h. For cytokine, PGE2, and nitrite quantification, media was collected after the treatment period for use in TNF-α, IL-6, or PGE2 ELISAs and in Griess reagent system nitrite assays. Cell viability was determined via MTS assay at this stage for inflammatory marker normalization. For COX2 Western blot studies, cell lysates were collected at the end of the treatment period, and COX2 bands were normalized to housekeeping protein GAPDH. Similar to prior experiments, drug and LPS doses were selected from prior pilot dose–response studies.

The incretin-based multi-agonists Twincretin, LY329, and Triagonist increased IMG cell viability by 43%, 54%, and 33%, respectively, in the presence of LPS (*p* < 0.0001), whereas exendin-4 produced no change in viability (Figure 5A). Twincretin, LY329, and Triagonist markedly reduced the LPS-induced production of the pro-inflammatory cytokines TNF-α (Figure 5B, *p* < 0.0001) and IL-6 (Figure 5C, *p* < 0.0001) relative to 1 ng/mL LPS alone. Exendin-4 reduced TNF-α (*p* < 0.05) and IL-6 (*p* < 0.001) levels relative to 1 ng/mL LPS to a lesser extent than the multi-agonists—Twincretin and Triagonist treatment reduced TNF-α levels (*p* < 0.001, *p* < 0.05) and IL-6 levels (*p* < 0.05, *p* = 0.0637) relative to the exendin-4 + LPS treatment group. All of the incretin-based peptides tested significantly reduced nitrite production by LPS-stimulated IMG cells (Figure 5D, exendin-4 *p* < 0.01, Twincretin *p* < 0.05, LY329 *p* < 0.0001, Triagonist *p* < 0.001), with LY329 levels significantly lower than exendin-4 (*p* < 0.0001), Twincretin (*p* < 0.0001), and Triagonist (*p* < 0.0001). LPS-induced COX2 expression (Figure 5E) was also significantly impeded by multi-agonist treatment (Twincretin *p* < 0.001, LY329 *p* < 0.0001, Triagonist *p* < 0.0001), with exendin-4 producing a less pronounced COX2 reduction (*p* < 0.05) compared to the multi-agonists. Moreover, cells treated with LY329 (*p* < 0.05) and Triagonist (*p* < 0.01) exhibited significantly reduced COX2 expression relative to those treated with exendin-4. We then quantified the concentration of PGE2, a downstream product of the COX2 signaling pathway, in cell culture media and found that exendin-4 (*p* < 0.01), Twincretin (*p* < 0.05), and LY329 (*p* < 0.001) significantly reduced PGE2 secretion from IMG cells relative to LPS alone (Figure 5F). The Triagonist treatment group trended towards a reduction in PGE2 relative to LPS alone, but this difference did not reach significance (Figure 5F, *p* = 0.2742). No significant differences in PGE2 levels between the incretin-based peptide treatment groups were observed. Notably, across the above studies (Figure 5B–F), data were normalized to cell viability (Figure 5A).

## 4. Discussion

The Cleveland Clinic estimates that over 50 million people across the globe are affected by neurodegenerative diseases, emphasizing the urgent need to develop effective treatments for neurodegeneration. A treatment approach, such as incretin mimetics, that can target multiple facets of neurodegenerative disease pathology, including neuroinflammation, IR, cytotoxicity, and oxidative stress, may be more effective than existing therapeutic options (Figure 6—generated from data derived from the scientific literature [61,62,63,64,65,66,67,68,69,70,71,72,73,74]). Our present and prior works have clearly indicated that incretin-based receptor agonists are effective in combatting neuroinflammation, oxidative stress, and apoptosis and offer neurotrophic and neuroprotective benefits [11,27,28,51,61,62,63]. In our current study, we hope to provide preclinical evidence that incretin-based multi-agonists may have enhanced success in mitigating neurodegenerative processes as compared to single GLP-1R agonists, and thus are promising candidates for repurposing in future neurodegenerative disease clinical trials.

To this end, we began by investigating the neurotrophic potential of two GLP-1R/GIPR dual agonists (Twincretin and LY329, available commercially for T2DM treatment as Tirzepatide/*Mounjaro*^®^) and a GLP-1R/GIPR/GcgR triple agonist (Triagonist) as compared to a single GLP-1R agonist, exendin-4. We demonstrated that all single, dual, and triple agonists tested enhanced the viability of neuronal SH-SY5Y cells (Figure 1A), reinforcing previous studies of multi-agonists’ neurotrophic potential presented by our research group [7,75]. Interestingly, our dual and triple agonists outperformed the single agonist exendin-4 in enhancing microglial viability in both human HMC3 and mouse IMG microglial cell lines (Figure 1B,C). Both GLP-1R and GIPR activity has been linked with trophic effects in microglia, including the upregulation of growth factors such as brain-derived neurotrophic factor (BDNF), glial cell line-derived neurotrophic factor (GDNF), and NGF, as well as the activation of antiapoptotic and antioxidative pathways [76]. Perhaps our observed microglial viability elevations with multi-agonists, relative to exendin-4, could be attributed to combined GLP-1R and GIPR signaling along neurotrophic pathways mediated by MAPK/ERK and mTOR (Figure 6), although synergistic trophic effects of joint GLP-1R/GIPR stimulation in microglia have not yet been observed [76], highlighting an avenue for further research.

Oxidative stress and ROS abundance drive neuronal and glial cell death, mitochondrial dysfunction, and inflammation in neurodegenerative disease [77,78], thus emphasizing the utility of drugs targeting oxidative stress to treat neurodegeneration. In our study, we found that incretin-based multi-agonists have significant neuroprotective and antioxidative capacity against a H_2_O_2_-induced oxidative stress challenge in both neuronal SH-SY5Y and microglial HMC3 cells. Of note, the dual and triple agonists demonstrated enhanced antioxidative and neuroprotective effects, as indicated by elevations in cell viability in the presence of H_2_O_2_ and reductions in cellular ROS relative to exendin-4, with some significant differences found between multi-agonist-treated groups and the exendin-4-treated group (Figure 2). Previous works in cellular and animal models of neurodegeneration have highlighted enhanced neuroprotective and antioxidative benefits of GLP-1R/GIPR dual agonists [75,79,80] and GLP-1R/GIPR/GcgR triple agonists [7,81] relative to single GLP-1R agonists, reinforcing our current findings. Mechanistically, downstream of GLP-1R/GIPR/GcgR activation, antioxidative signaling driven by Epac and APE1 may be involved in protecting cells and organisms against oxidative stress and oxidative stress-induced neurodegeneration (Figure 6).

Caspase-3-dependent and -independent mechanisms of apoptosis additionally play a role in neuronal death in neurodegenerative disease. In our experiments, we utilized an established pro-apoptotic agent, STS, to stimulate apoptosis in neuronal SH-SY5Y cells and evaluated whether our incretin-based dual and triple agonists could protect cells against apoptosis. We quantified cytotoxicity following treatment and observed that the single agonist exendin-4, the dual agonist Twincretin, and the triple agonist Triagonist reduced SH-SY5Y cytotoxicity induced by 100 nM and/or 250 nM STS with increasing degrees of significance (Figure 3). Interestingly, in a parallel experiment measuring caspase-3 activity, only Triagonist significantly reduced caspase-3 activity (Figure 4), suggesting that STS and incretin-based peptides may interact with apoptotic pathways through both caspase-3-dependent and -independent mechanisms, perhaps involving the Bcl-2/Bcl-xL-mediated inhibition of apoptotic executioners and/or the Akt-mediated inhibition of pro-apoptotic proteins (Figure 6). Indeed, other groups have found that dual GLP-1R/GIPR agonists reduced the Bax (pro-apoptotic) to Bcl2 (antiapoptotic) ratio as well as increasing the levels of phosphorylated Akt in cortical and hippocampal cells derived from rat stroke and AD models [82,83], with the dual agonist providing greater antiapoptotic protections than a GLP-1R single agonist [82]. Future research could investigate the relative contributions of various antiapoptotic signaling pathways implicated in GLP-1R/GIPR/GcgR antiapoptotic activity, as well as the relative contributions of GLP-1R, GIPR, and GcgR stimulation in mitigating apoptosis.

The cycle of chronic neuroinflammation in neurodegenerative disease is an attractive treatment target for anti-neurodegenerative drug development (Figure 7). Microglial activation into a pro-inflammatory phenotype can occur following a multitude of insults relevant to various neurodegenerative diseases, including TBI, infection, excess glucose, pathogens, misfolded amyloid-β or α-synuclein, oxidative stress, and neuronal cell debris [46] (Figure 7A,B,F). Pro-inflammatory microglia secrete inflammatory factors including pro-inflammatory cytokines (such as TNF-α, IL-6, and IL-1β), chemokines, nitric oxide (NO), PGE2, and ROS, which can directly damage neurons (Figure 7D,E) and activate additional pro-inflammatory microglia (Figure 7C), and also promote astrocytic conversion into a reactive neurotoxic phenotype [6,26,46,84]. Reactive astrocytes release neurotoxic saturated lipids in apolipoprotein (APO)E and APOJ lipoparticles and thus drive further neurodegeneration [85]. Incretins can act to combat neuroinflammation through the Akt-mediated inhibition of NF-κB (Figure 6), a driver of microglial phenotypic conversion into a pro-inflammatory state [86], pro-inflammatory cytokine release [87], COX2 expression, and downstream PGE2 release [88].

Thus, in our inflammation experiments, we chose to quantify the secreted levels of two pro-inflammatory cytokines (TNF-α and IL-6), nitrite, and PGE2, as well as intracellular COX2 protein levels, an enzyme upstream of prostaglandin production with known roles in inflammatory processes [89]. In these experiments, we pre-treated IMG microglial cells with exendin-4, Twincretin, LY329, or Triagonist, then stimulated the cells to adopt a pro-inflammatory phenotype using LPS. We noted a marked difference between multi-agonist and single GLP-1R agonist treatment in the majority of these inflammatory assays (which were normalized to cell viability (Figure 5A)). Twincretin, LY329, and Triagonist reduced LPS-induced TNF-α secretion by approximately 30–49%, compared to about 16% achieved with exendin-4 (Figure 5B). Similarly, all three multi-agonists reduced LPS-induced IL-6 secretion by roughly 37–50%, as opposed to 25% achieved with exendin-4 (Figure 5C). We noticed fewer differences between single and multi-agonists in terms of LPS-induced nitrite and PGE2 production—all incretin peptides significantly reduced nitrite and PGE2 levels, with the exceptions of LY329 producing a notable 47% reduction in nitrite (Figure 5D) and an insignificant effect of Triagonist on PGE2 levels (Figure 5F). As for COX2 levels, Twincretin, LY329, and Triagonist produced 26%, 36%, and 37% reductions in COX2 expression, respectively, compared with only a 16% reduction produced by exendin-4 (Figure 5E). In addition, microglial viability in the presence of LPS was increased by an impressive 43% with Twincretin, 54% with LY329, and 33% with Triagonist, with little to no effect on viability with exendin-4 (Figure 5A).

It has been well established in mammalian cells that LPS induces inflammation primarily through TLR4 stimulation. A recent paper demonstrated that central neuronal GLP-1R activity plays an essential role in combatting TLR4-mediated inflammation [90]. Further, the anti-inflammatory effects of single GLP-1R agonist exendin-4 have been previously reported in RAW264.7 macrophages challenged with LPS [59]. Here, we demonstrate for the first time that in microglial cells, in addition to exendin-4, incretin-based multi-agonists can significantly reduce multiple LPS-induced inflammation mediators, such as nitrite, COX2, PGE2, and pro-inflammatory cytokines TNF-α and IL-6, in line with previous findings [59].

Our results in IMG cells indicate that multi-agonists could elicit anti-inflammatory effects directly through GLP-1R activity, as well as through GIPR or GcgR activity, in microglia. Although most existing studies of the neuroprotective and anti-inflammatory activity of incretin-based compounds focus on GLP-1R-mediated benefits, we and others have shown that the activation of GIPR has similar effects [91,92]. In addition, our previous study of Triagonist in SH-SY5Y cells has shown that each receptor (GLP-1R, GIPR, and GcgR) contributed to the drug’s demonstrated neuroprotective effects [7], further emphasizing the possible advantages of multi-agonism over single GLP-1R agonism in the treatment of various neurological disorders.

Our results reproduce the superior anti-inflammatory effects of incretin-based multi-agonists compared to single GLP-1R agonists tested in previous preclinical studies. In a mouse model of PD, a novel GLP-1R/GIPR dual agonist outperformed single GLP-1R agonist NLY01 in reducing the levels of activated NF-κB, TNF-α, IL-6, and IL-1β, as well as mitigating microglial activation and reactive astrogliosis [93]. Further, a mouse study of AD found that treatment with a dual GLP-1R/GIPR agonist significantly reduced reactive astrogliosis relative to treatment with the single GLP-1R agonist liraglutide [80]. Our group has previously demonstrated Triagonist to induce cAMP significantly more than exendin-4 in SH-SY5Y cells, and to reduce TNF-α against an LPS challenge in primary mouse microglia [7]. Similarly, a triple GLP-1R/GIPR/GcgR agonist was found to impede reactive astrogliosis and microglial pro-inflammatory activation in a mouse model of AD [81]. Through reducing glial cell inflammatory activity and enhancing glial viability, incretin-based multi-agonists may allow for reparative functions of glial cells to combat neuroinflammation and neurodegeneration.

The benefit of incretin-based multi-agonists as treatments for neurodegenerative disease lies not only in their pleiotropic therapeutic mechanisms, but also in the relative efficiency of repurposing an existing FDA-approved drug class, as compared to developing entirely new therapeutics. The safety profiles of GLP-1R single agonists as well as dual GLP-1R/GIPR agonists, namely Tirzepatide/*Mounjaro*^®^ (LY329), and triple GLP-1R/GIPR/GcgR agonists, namely Retatrutide (Triagonist), have already been established in humans, thereby saving significant amounts of time and resources. Considering the urgency of the demand for neurodegenerative disease treatments, the efficiency that repurposing incretin mimetics offers is immensely valuable.

However, an important consideration with repurposing incretin-based compounds to treat diseases of the brain is drug administration. GLP-1, GIP, and Gcg are large peptide compounds, and synthetic versions of these compounds with additional chemical modifications can be even larger. As a result, limited BBB penetration is an area of concern with this class of drugs, particularly considering that the drugs are administered peripherally [94,95]. Further research that investigates mechanisms of drug entry into the brain as well as pharmacokinetic studies with the more novel incretin-based multi-agonists will certainly be valuable in identifying translatable doses relevant to human neurodegenerative disease treatment.

The development and emergence of incretin-based multi-agonists on the market for T2DM and obesity treatment are quite encouraging. Although Tirzepatide/*Mounjaro*^®^ is the only currently FDA-approved multi-agonist, several additional incretin-based dual or triple agonists are in clinical trials for T2DM and obesity, including the triple agonist (Retatrutide) used in this study [96]. Tirzepatide/*Mounjaro*^®^ has already demonstrated enhanced efficacy in T2DM treatment relative to single GLP-1R agonists [48]. In much the same way, in our present study, multi-agonists magnified anti-neuroinflammatory and neuroprotective benefits as compared to a single GLP-1R agonist in cell culture models of neurodegenerative disease. In this regard, three different immortal CNS cell lines were evaluated in the present study to determine whether incretin-mediated actions translate across different cell types (neuronal vs. microglial), as well as across rodent and human derived cells (IMG and HMC3 microglia, respectively). Future in vivo studies and clinical trials are certainly warranted to determine whether incretin receptor multi-agonists may outperform single GLP-1R agonists as a novel therapeutic option for human neurodegenerative diseases.

## Figures and Tables

**Figure 1 biomolecules-14-00872-f001:**
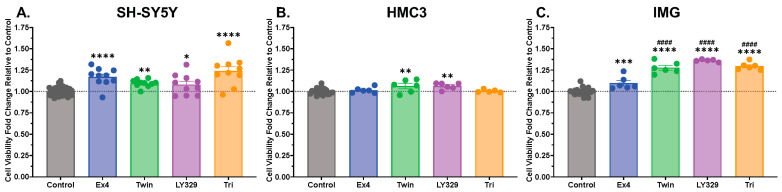
Neurotrophic effects of exendin-4 (Ex4) and multi-agonists Twincretin (Twin), LY329, and Triagonist (Tri), presented as cell viability fold changes relative to the Control condition. SH-SY5Y human neuronal cells (**A**) or HMC3 human microglial cells (**B**) were incubated with the vehicle (SH-SY5Y n = 48, HMC3 n = 23) or 10 nM exendin-4, Twincretin, LY329, or Triagonist (SH-SY5Y n = 10, HMC3 n = 5–6 per treatment group) for 48 h; mouse IMG cells (**C**) were incubated with the vehicle (n = 20) or 100 nM exendin-4, Twincretin, LY329, or Triagonist (n = 5–6 per treatment group) for 48 h. Cell viability was quantified using an MTS assay as per kit instructions. Values are expressed relative to Control levels. * indicates a significant difference at *p* < 0.05, ** at *p* < 0.01, *** at *p* < 0.001, **** at *p* < 0.0001 from Control; #### indicates a significant difference at *p* < 0.0001 from Ex4 displayed by one-way ANOVA with Dunnett’s multiple comparisons test.

**Figure 2 biomolecules-14-00872-f002:**
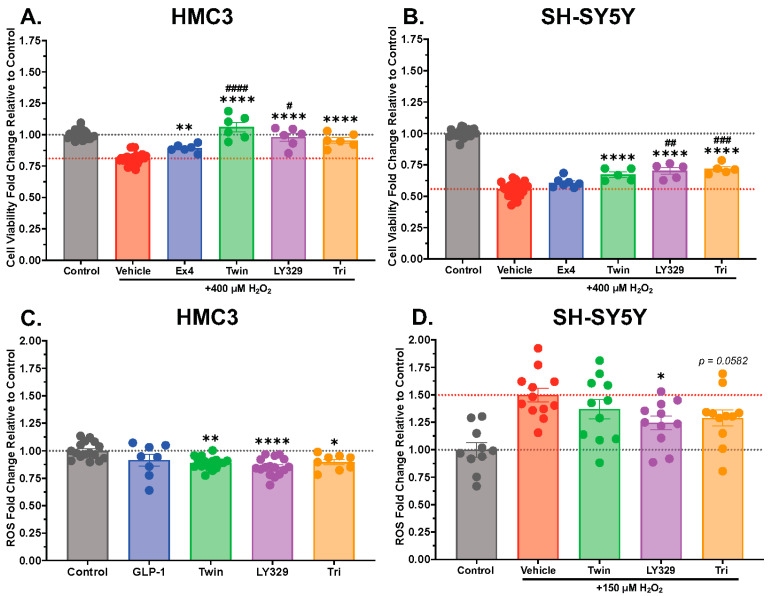
Antioxidative stress and neuroprotective signaling of incretin-based multi-agonists in HMC3 and SH-SY5Y cells, presented as fold changes in cell viability (**A**,**B**) or ROS levels (**C**,**D**) relative to the Control condition. In cell viability experiments, HMC3 human microglial cells (**A**) or SH-SY5Y human neuronal cells (**B**) were pre-treated with the vehicle (HMC3 Control n = 23, HMC3 Vehicle + H_2_O_2_ n = 24, SH-SY5Y Control n = 22, SH-SY5Y Vehicle + H_2_O_2_ n = 22) or 10 nM exendin-4, Twincretin, LY329, or Triagonist (HMC3 n = 6, SH-SY5Y n = 5–7 per treatment group) for 24 h, followed by a 24 h challenge treatment with 400 µM H_2_O_2_. Cell viability was quantified using an MTS assay as per kit instructions (**A**,**B**). In ROS quantification experiments, ROS levels were measured by DCFDA assays in HMC3 (**C**) and SH-SY5Y cells (**D**). Cells were pre-treated for 2 h with vehicle (HMC3 Control n = 16, SH-SY5Y Control n = 10, SH-SY5Y Vehicle + H_2_O_2_ n = 12) or 100 nM GLP-1 (HMC3 only, n = 8), Twincretin (HMC3 n = 16, SH-SY5Y n = 11), LY329 (HMC3 n = 16, SH-SY5Y n = 11), or Triagonist (HMC3 n = 8, SH-SY5Y n = 11), followed by a 1 h challenge with 150 µM H_2_O_2_. HMC3 cells demonstrated basal ROS reductions with Twincretin, LY329, and Triagonist treatment (**C**), but no significant ROS-reducing incretin effects were observed in the presence of the H_2_O_2_ challenge. In contrast, in SH-SY5Y cells, LY329 and Triagonist treatments produced significant or near-significant ROS reductions with the H_2_O_2_ challenge (**D**). ROS levels were determined using a DCFDA assay as per the kit instructions (**C**,**D**). All values are expressed relative to Control levels. * indicates a significant difference at *p* < 0.05, ** at *p* < 0.01, **** at *p* < 0.0001 from Vehicle + H_2_O_2_ (**A**,**B**,**D**) or from Control (**C**); # indicates a significant difference at *p* < 0.05, ## at *p* < 0.01, ### at *p* < 0.001, #### at *p* < 0.0001 from Ex4 + H_2_O_2_ by one-way ANOVA with Dunnett’s multiple comparisons test.

**Figure 3 biomolecules-14-00872-f003:**
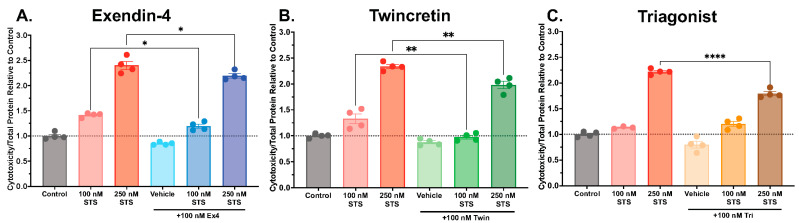
Antiapoptotic effects of incretin-based single, dual, and triple receptor agonists in neuronal cells. SH-SY5Y cells were pre-treated for 24 h with vehicle or 100 nM exendin-4 (**A**), Twincretin (**B**), or Triagonist (**C**), followed by a 24 h 100 or 250 nM STS pro-apoptotic challenge treatment (n = 3–4 per treatment group). Cytotoxicity levels were quantified using an LDH assay as per the kit instructions and were normalized to total protein level as determined by BCA. Values are expressed relative to Control levels. * indicates a significant difference between STS alone and STS + agonist treatment at *p* < 0.05, ** at *p* < 0.01, **** at *p* < 0.0001 by one-way ANOVA with Tukey’s multiple comparisons test.

**Figure 4 biomolecules-14-00872-f004:**
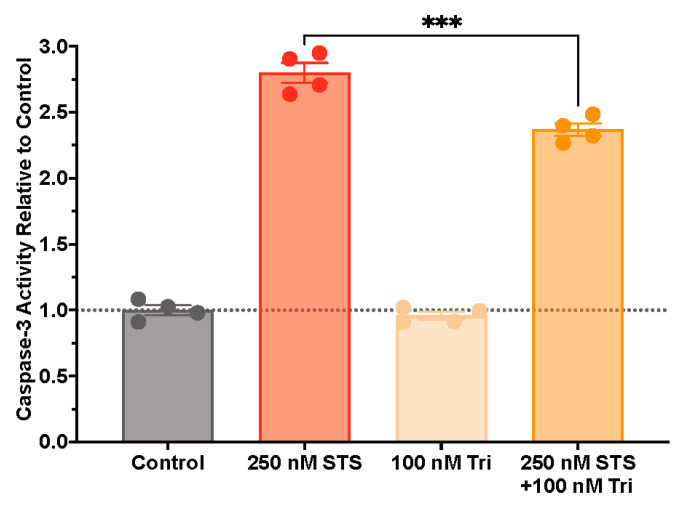
Pro-apoptotic caspase-3 activity is reduced with Triagonist treatment. SH-SY5Y cells were pre-treated with the vehicle or 100 nM Triagonist for 24 h; 250 nM STS was subsequently added for an additional 24 h (n = 4 per treatment group). Caspase-3 activity was determined using a Fluorimetric Caspase 3 Assay Kit as per the kit instructions. Of note, Triagonist was the only incretin-based compound that we found to be effective in mitigating caspase-3 activity. Values are expressed relative to Control levels. *** indicates a significant difference at *p* < 0.001 by one-way ANOVA with Tukey’s multiple comparisons test.

**Figure 5 biomolecules-14-00872-f005:**
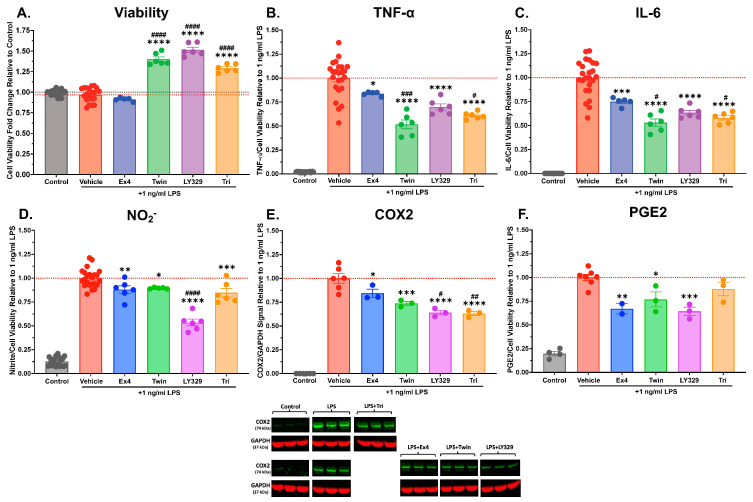
Wide-ranging anti-neuroinflammatory abilities of incretin receptor single and multi-agonists. (**A**) Multi-agonists enhance IMG cell viability and, notably, reduce pro-inflammatory microglial signaling, nitrite production, and cytokine release (following normalization to cell viability). TNF-α (**B**), IL-6 (**C**), and PGE2 (**F**) levels in culture media were determined using corresponding ELISA kits; nitrite (NO_2_^−^) levels (**D**) in culture media were determined using a Griess reagent system nitrite assay kit; and intracellular COX2 expression (**E**) was determined using Western blotting and was normalized to GAPDH expression (representative bands are shown in (**E**) (bottom), all blots are provided in the Appendix A). Mouse IMG cells were pre-treated with 100 nM exendin-4, Twincretin, LY329, or Triagonist for 24 h, followed by a 24 h 1 ng/mL LPS pro-inflammatory treatment. Cytokine, nitrite, and PGE2 levels were normalized to cell viability as determined via MTS assays (**A**). In (**A**), values are expressed relative to Control levels; in (**B**–**F**), values are expressed relative to 1 ng/mL LPS levels. (**A**–**D**) Control n = 19–23, Vehicle + LPS n = 20–24, each peptide + LPS n = 5–6; (**E**) Control n = 6, Vehicle + LPS n = 6, each peptide + LPS n = 3; (**F**) Control n = 4, Vehicle + LPS n = 7, each peptide + LPS n = 2–3. * indicates a significant difference at *p* < 0.05, ** at *p* < 0.01, *** at *p* < 0.001, **** at *p* < 0.0001 from 1 ng/mL LPS; # indicates a significant difference at *p* < 0.05, ## at *p* < 0.01, ### at *p* < 0.001, #### at *p* < 0.0001 from Ex4 + LPS according to one-way ANOVA with Dunnett’s multiple comparisons test. Original Western blot images are available in Appendix A.

**Figure 6 biomolecules-14-00872-f006:**
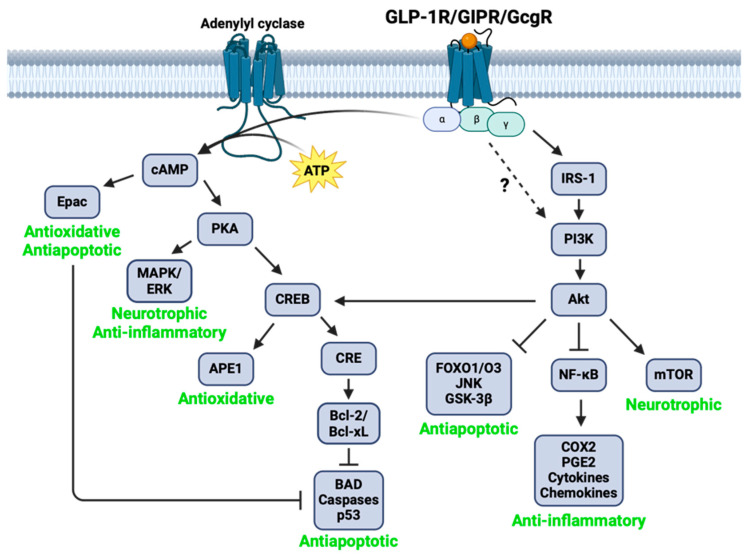
Molecular mechanisms of anti-neurodegenerative action of incretin mimetics in the CNS (supported by the scientific literature). Upon ligand binding, the alpha subunit of GLP-1R/GIPR/GcgR activates adenylyl cyclase and drives cyclic adenosine monophosphate (cAMP) production. In turn, cAMP activates exchange protein activated by cAMP (Epac), involved in antioxidant generation and antiapoptotic signaling [64,65], as well as protein kinase A (PKA). Downstream of PKA activity, mitogen-activated protein kinase (MAPK)/extracellular signal-regulated kinase (ERK) signaling promotes cell survival and proliferation [66], and cAMP-response element (CRE) binding protein (CREB) enhances the expression of B-cell lymphoma (Bcl)-2 and extra-large (Bcl-xL), inhibitors of the apoptotic executioners BCL2-associated agonist of cell death (BAD), caspases, and p53 [67,68,69,70]. CREB also stimulates apurinic/apyrimidinic endonuclease 1 (APE1), with roles in repairing DNA damage resulting from oxidative stress [71,72]. Activated GLP-1R/GIPR/GcgR additionally stimulates insulin receptor substrate (IRS)-1 and phosphoinositide 3-kinase (PI3K), thus activating Ak strain transforming (Akt) and driving anti-inflammatory actions through the inhibition of nuclear factor kappa-light-chain-enhancer of activated B cells (NF-κB), antiapoptotic actions through inhibition of forkhead box (FOX)O1/O3, c-Jun N-terminal kinase (JNK), and glycogen synthase kinase (GSK)-3β [70], and trophic actions through mammalian target of rapamycin (mTOR) signaling [73,74]. Figure adapted from Glotfelty et al., 2019 [6].

**Figure 7 biomolecules-14-00872-f007:**
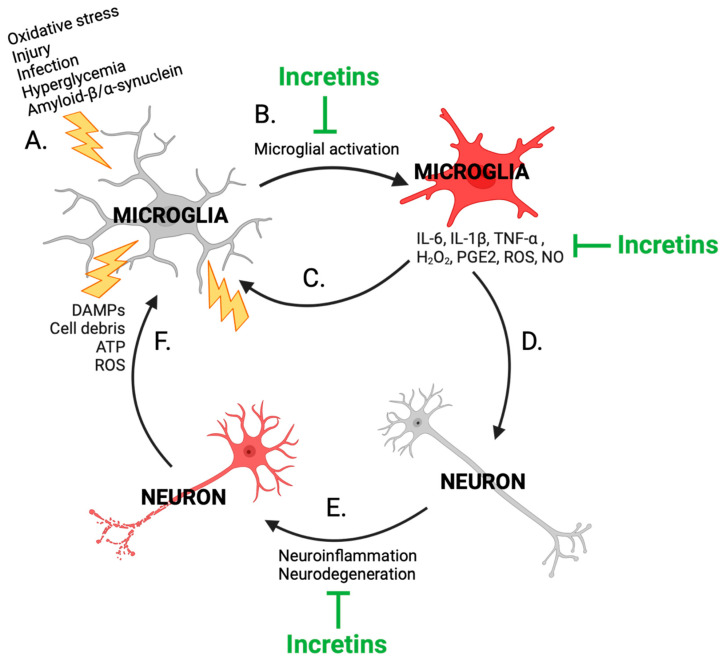
Cycle of neuroinflammation and sites of incretin-based compound anti-inflammatory activity (generated from the present studies and the scientific literature [46]). CNS stressors relevant to neurodegenerative disease (**A**) can activate microglia into a pro-inflammatory phenotype (**B**). Activated microglia release various pro-inflammatory factors, including cytokines, chemokines, prostaglandins, and other inflammatory molecules, that can bind neurons (**D**) and activate additional microglia (**C**). Damaged and degenerating neurons (**E**) secrete signaling molecules such as damage-associated molecular patterns (DAMPs), cell debris, ATP, and ROS that in turn activate microglia (**F**), fueling the neuroinflammatory cycle. Incretin-based peptides (green) can act at several of these stages to mitigate neuroinflammation—incretins can dampen microglial activation, reduce pro-inflammatory cytokine release, mitigate oxidative stress, and promote neuronal viability. Figure adapted from Kopp et al., 2022 [46].

## Data Availability

Data will be provided on reasonable request to the authors.

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
