# Peer review of "Incretin-Based Multi-Agonist Peptides Are Neuroprotective and Anti-Inflammatory in Cellular Models of Neurodegeneration"

_biomolecules, 2024, doi:10.3390/biom14070872_

Round 1

Reviewer 1 Report

Comments and Suggestions for Authors

Strength:

The authors aimed to elucidate whether multiple incretin-related receptor agonists have neuroprotective and anti-inflammatory effects in cellular models of neurodegeneration using HMC3, IMG and SH-SY5Y. The authors demonstrated that two selected GLP-1R/GIPR dual agonists and a GLP-1R/GIPR/GcgR triple agonist not only have neurotrophic and neuroprotective effects, but also anti-neuroinflammatory effects. These results suggest that these multiple incretin-related receptor agonists have potential in the treatment of neurodegenerative diseases.

Major comments :

1In all experiments, the pharmacological effects of multiple incretin-related receptor agonists should be shown to be concentration dependent. It is not appropriate to infer pharmacological properties from the result obtained with a single concentration of agonist.

2Both HMC3 and SH-SY5Y were derived from human tissue. In contrast, IMG was established from mouse tissue. Authors must consider the species difference in the character of these cell lines. It is inappropriate to treat experimental results obtained from different cell types from different species as equivalent.  

3. In this study, three materials, namely H2O2, sutaurosporine and lipopolysaccharide, were used as reagents to induce a neurodegenerative effect for HMC3, SH-SY5Y and IMG. However, the experimental combinations between three cell lines and three materials are inconsistent.  For example, the effect of H2O2 on IMG was not tested, although other cell lines including HMC-3 and SH-SY5Y were tested in Figure 2. Figure 3 also shows that the antiapoptotic effect induced by staurosporine was only tested in SH-SY5Y but not in HMC3 and IMG.

4. In Figure 5, the authors should perform the same experiments using HMC-3. It is inappropriate to conclude the pharmacological effects of multiple incretin-related receptor agonists by integrating the experimental results of two microglial cell lines established in different species.

5. Figures 6 and 7 seem excessive, as most of the experimental data do not support the content of these figures. For example, it is not clear that incretins can inhibit microglial activation induced by amyloid-β/α-synuclein in this experimental condition in Figure 7.  In Figure 6, the authors also failed to show how multiple incretin-related receptor agonists affected different internal cell signalling.  Authors should only express what they have done.

Minor comments :

1. The authors need to show the results of Western blotting for COX2 and GAPDH in Figure 5E.

2. In Figure 5A, please discuss why cell viability was increased by treatment of Twincretin, LY329 and Triagonist with lipopolysaccharide. In addition, please also consider how the increase in cell viability affects the secretion of various cytokines, NO production and COX2 and PGE2 expression.

Author Response

1.In all experiments, the pharmacological effects of multiple incretin-related receptor agonists should be shown to be concentration dependent. It is not appropriate to infer pharmacological properties from the result obtained with a single concentration of agonist.

Response: We thank the Reviewer for making this important point – which we fully agree with. We, indeed, performed dose-response experiments for all of our peptides in all 3 cell lines used in the manuscript, and found that peptide concentrations between 10-100 nM demonstrated optimal effects. In contrast, concentrations below 10 nM usually showed no effects. In our studies, we therefore selected ‘threshold” concentrations of incretins that induced a biological action across the peptides studied – that were chosen from our pilot dose-response experiments. This is specifically noted in the 1st paragraph of Page 7 within the manuscript (Results section) – where we state: “… These incretin-based peptide concentrations were specifically selected from pilot study dose-response curves in each cell line focused to define the threshold activity of the peptides.”

We noted this important information in relation to the selection of our peptide dose, rather than show the preliminary does-response results in the Figures due to the significant space that these preliminary results would have taken. (Please note that we routinely undertake ‘dose-response’ evaluations in our studies – see, for example, Li et al., J. Neurochem 2021 < https://www.ncbi.nlm.nih.gov/pmc/articles/pmid/34569615/> ).

Also please note, that we likewise state that incretin dose selection in other Figures derives from pilot dose-response studies performed to define a ‘threshold dose’ to provide a biological action (see page 8).

Finally, in the light of the concern of Reviewer 1 in relation to dose selection, we have added the following short paragraph to the ‘Methods section’:

Drug and Challenge Dose Selection: The selection of an incretin mimetic dose to evaluate in neurotrophic, neuroprotective and inflammatory studies across SH-SY5Y, HMC3 and IMG cells was based on pilot dose-response studies that were focused on identifying the threshold concentration to provide the biological action of interest, and this dose was then used across single, dual and triple receptor agonist to allow their side-by-side comparison at an equimolar concentration. Likewise, for the selection of a physiological/drug-induced (i.e., H2O2, LPS, STS) challenge, concentrations were selected from prior pilot dose-response studies focused to achieve a biologically significant but sub-maximal effect.”

2.Both HMC3 and SH-SY5Y were derived from human tissue. In contrast, IMG was established from mouse tissue. Authors must consider the species difference in the character of these cell lines. It is inappropriate to treat experimental results obtained from different cell types from different species as equivalent.  

Response: We agree with Reviewer 1 that the three cell lines we used are derived from both human and mouse, and that species difference should be considered. In addition, each cell line has its own characteristics. For example, HMC3s and IMG cells are both microglia cell lines derived from human and mouse cells respectively, but IMG cells are more sensitive to LPS-induced inflammation (in terms of cytokine release), whereas HMC3 cells only have detectable IL-6 in the media but not TNF-a. We observed a trend of decreased IL-6 levels by multi-agonists in the LPS experiment in HMC3 cells (not shown in the figures) and have added this result  to the written content.). While cell lines are not perfect models, using multiple species and evaluating ‘big picture’ trends across these makes for more convincing results. It should be noted that there are limited well-characterized human microglial cell line models available and our previous work (see Glotfelty et al., 2023) has shown IMG cells to be a good model for primary mouse microglia.

It should also be noted that it is not unusual to see data relating from cell lines from different species within a single scientific publication – particularly related to drug studies. This provides support that drug actions are not necessarily species specific, and that preclinical cellular studies relating to, for example, rodent potentially have relevance to human studies.    

We added the following sentence to the final paragraph of the Discussion: “….In this regard, three different immortal neural cell lines were evaluated in the present study, to determine whether incretin-mediated actions translate across different cell types (neuronal vs. microglial), as well as across rodent and human derived cells (IMG and HMC3 microglia, respectively).”

3. In this study, three materials, namely H2O2, sutaurosporine and lipopolysaccharide, were used as reagents to induce a neurodegenerative effect for HMC3, SH-SY5Y and IMG. However, the experimental combinations between three cell lines and three materials are inconsistent.  For example, the effect of H2O2 on IMG was not tested, although other cell lines including HMC-3 and SH-SY5Y were tested in Figure 2. Figure 3 also shows that the antiapoptotic effect induced by staurosporine was only tested in SH-SY5Y but not in HMC3 and IMG.

Response: We understand the Reviewer’s concerns and have tried to include all 3 cell lines in most experiments. However, due to the different characteristics of the cell lines, for example, IMG cells attach the surface loosely, and therefore come off easily during assay procedures, such cells are not compatible with evaluating particular challenges – for example, the ROS assay protocols we use.)

4. In Figure 5, the authors should perform the same experiments using HMC-3. It is inappropriate to conclude the pharmacological effects of multiple incretin-related receptor agonists by integrating the experimental results of two microglial cell lines established in different species.

Response: Please see our responses to Reviewer comment #2 and #3)

5. Figures 6 and 7 seem excessive, as most of the experimental data do not support the content of these figures. For example, it is not clear that incretins can inhibit microglial activation induced by amyloid-β/α-synuclein in this experimental condition in Figure 7.  In Figure 6, the authors also failed to show how multiple incretin-related receptor agonists affected different internal cell signalling.  Authors should only express what they have done.

Response: We understand the viewpoint of Reviewer 1.  However, in contrast, Reviewer 2 very much liked both figures 6 and 7 (noting that they are ‘impressive’). We specifically placed Figures 6 and 7 within the Discussion section (not in the Results section) so that they can be used as a prospective in the Discussion (even though they are not specifically supported by the data generated and presented in the manuscript). To make this absolutely clear, we have added to the Figure legends the phrase “(supported by the scientific literature)” – to indicate that the pathway data derives from the published literature (rather than from the present study results) – and scientific references are noted in support of the pathway information

Minor comments :

1. The authors need to show the results of Western blotting for COX2 and GAPDH in Figure 5E.

Response: We fully agree with the Reviewer, and have added representative Western blots. We thank the Reviewer for the suggestion. (All Western blots have been provided to the journal as 'Supplemental Information' - and have been uploaded via their website).

2. In Figure 5A, please discuss why cell viability was increased by treatment of Twincretin, LY329 and Triagonist with lipopolysaccharide. In addition, please also consider how the increase in cell viability affects the secretion of various cytokines, NO production and COX2 and PGE2 expression.

Response: In the IMG cells in Fig. 5A, cell viability was increased to a similar extent with incretins added in the ABSENCE of LPS (see Fig. 1C) as well as in the presence (Fig. 5A). The increase in viability hence doesn’t appear to be LPS-dependent, and an explanation of why the multi-agonists increased viability relative to exendin-4 is provided in the Discussion: “... Perhaps our observed microglial viability elevations with multi-agonists, relative to exendin-4, could be attributed to combined GLP-1R and GIPR signaling along neurotrophic pathways mediated by MAPK/ERK and mTOR (Figure 6), albeit synergistic trophic effects of joint GLP-1R/GIPR stimulation in microglia have not yet been observed [76] , highlighting an avenue for further research.”

To address the Reviewer’s important second point, the data shown in Fig. 5 B,C,D,E,F is “normalized” to cell viability (noted on the axis) and the graphs are thus indicative of cytokine/PGE2/COX2 levels “per cell”. This important ‘normalization’ is also noted in the legend to Fig. 5 (“Cytokine, nitrite, and PGE2 levels were normalized to cell viability as determined via MTS assay (A)”).

Reviewer 2 Report

Comments and Suggestions for Authors

The authors aimed to evaluation of novel incretin receptor multi-agonists in preclinical models of neurodegeneration and  thereby inform potential future clinical trials.

The results of this study presented in Figures very clearly.

Figure 6 and 7 is very impresive.

The authors highlight promising findings in cell culture models of neurodegeneration indicating enhanced neuroprotective, antioxidative, and anti-inflammatory benefits of three select multi-agonists—dual GLP-1R/GIPR agonists “Twincretin” and LY329  and triple GLP-1R/GIPR/GcgR agonist “Triagonist” relative to  single GLP-1R agonist, exendin-4.

Comment

Write the literature in the appropriate font.

Author Response

The authors aimed to evaluation of novel incretin receptor multi-agonists in preclinical models of neurodegeneration and  thereby inform potential future clinical trials.

The results of this study presented in Figures very clearly.

Figure 6 and 7 is very impressive.

The authors highlight promising findings in cell culture models of neurodegeneration indicating enhanced neuroprotective, antioxidative, and anti-inflammatory benefits of three select multi-agonists—dual GLP-1R/GIPR agonists “Twincretin” and LY329  and triple GLP-1R/GIPR/GcgR agonist “Triagonist” relative to  single GLP-1R agonist, exendin-4.

Comment

Write the literature in the appropriate font.

Response: We agree with the Reviewer and have changed the font according to the instruction. Thank you for pointing this out.

Reviewer 3 Report

Comments and Suggestions for Authors

The article is well-structured with clear sections including Materials and Methods, Results, and Figures. The writing is generally clear and technical, appropriate for a scientific audience familiar with cellular assays and peptide pharmacology. However, a few areas could benefit from minor adjustments for clarity. Materials enlists all the reagents, including peptides, chemicals, assay kits, and Cell Culture protocols ensuring reproducibility.

Ensure that each figure legend clearly explains the experimental setup, treatment conditions, and statistical significance. Using graphs (Figure 1A, B, C) to illustrate the viability of SH-SY5Y, HMC3, and IMG cells after treatment with different peptides is effective. However, ensure that the legends and axis labels are clear and properly explained in the text for accurate data interpretation. The legend explanations of figures 2A, B, C, and D could be more detailed to match the complexity of the data shown. In Figures 3A, B, and C the legend should specify the statistical significance clearly to aid interpretation.

Mention the specific statistical tests used for each analysis in the main text. For example, state if ANOVA or t-tests were applied and the significance thresholds. Double-check the consistency in formatting, especially with units (e.g., µM, nM) and abbreviations (e.g., ELISA, ROS). Generally, the language is technical and appropriate for a scientific manuscript. Ensure that sentence structures are clear and easy to follow throughout.

Overall, the article provides a detailed account of the experimental setup and results concerning the neuroprotective and anti-inflammatory effects of incretin-based multi-agonist peptides in neurodegenerative cellular models. With minor adjustments for clarity in figure legends and statistical details, the manuscript should effectively communicate its findings to the scientific community.

Comments on the Quality of English Language

Minor editing of the English language required

Author Response

Comments and Suggestions for Authors

The article is well-structured with clear sections including Materials and Methods, Results, and Figures. The writing is generally clear and technical, appropriate for a scientific audience familiar with cellular assays and peptide pharmacology. However, a few areas could benefit from minor adjustments for clarity. Materials enlists all the reagents, including peptides, chemicals, assay kits, and Cell Culture protocols ensuring reproducibility. 

Ensure that each figure legend clearly explains the experimental setup, treatment conditions, and statistical significance. Using graphs (Figure 1A, B, C) to illustrate the viability of SH-SY5Y, HMC3, and IMG cells after treatment with different peptides is effective. However, ensure that the legends and axis labels are clear and properly explained in the text for accurate data interpretation. The legend explanations of figures 2A, B, C, and D could be more detailed to match the complexity of the data shown. In Figures 3A, B, and C the legend should specify the statistical significance clearly to aid interpretation.

Response: We agree with the Reviewer and have made revisions to the Figure legends in the light of the above comments

Mention the specific statistical tests used for each analysis in the main text. For example, state if ANOVA or t-tests were applied and the significance thresholds. Double-check the consistency in formatting, especially with units (e.g., µM, nM) and abbreviations (e.g., ELISA, ROS). Generally, the language is technical and appropriate for a scientific manuscript.

Response: We agree with the Reviewer and have added additional information on statistical analyses performed on the data sets (both in the Materials and Methods section (under ’Statistical Analysis’), as well as within the Figure legends.

Ensure that sentence structures are clear and easy to follow throughout.

Overall, the article provides a detailed account of the experimental setup and results concerning the neuroprotective and anti-inflammatory effects of incretin-based multi-agonist peptides in neurodegenerative cellular models. With minor adjustments for clarity in figure legends and statistical details, the manuscript should effectively communicate its findings to the scientific community.

Response: We have made minor edits throughout our revised manuscript to both clarify and improve the flow of English to improve our written communication

Round 2

Reviewer 1 Report

Comments and Suggestions for Authors

Comments:

The authors have responded appropriately to several major comments #1 and all minor comments. However, they have not yet addressed my concerns about the other major comments.